# IgG4-Related Oesophageal Disease with Cytomegalovirus Infection: A Case Report

**DOI:** 10.3390/jpm13030493

**Published:** 2023-03-09

**Authors:** Bacui Zhang, Yuexing Lai, Yongwei Xu, Jing Wang, Ping Xu

**Affiliations:** Department of Gastroenterology, Songjiang Hospital Affiliated to Shanghai Jiaotong University School of Medicine (Preparatory Stage), Shanghai 201600, China

**Keywords:** IgG4, IgG4-related disease, oesophageal ulcer, cytomegalovirus, case report

## Abstract

Immunoglobulin G4-related disease (IgG4-RD) is a fibrous inflammatory process related to immunomodulation. The involvement of the pancreato-biliary tract, retroperitoneum/aorta, head and neck, and salivary glands are the most frequently observed disease phenotypes, differing in their epidemiological features, serological findings, and prognostic outcomes. IgG4-RD was combined with oesophageal ulcers, and the patients were infected with cytomegalovirus at the time of the examination. This constituted a huge challenge in the diagnosis and treatment of oesophageal ulcers. We report the case of a 53-year-old male who experienced nausea, vomiting, and anaemia recurrently for many years. According to his medical records, an upper gastrointestinal endoscopy revealed an oesophageal ulcer, and he had had numerous hospital visits for anaemia but with no definitive diagnosis, and he had responded poorly to therapy. However, with persistent symptoms, he came to our hospital and, according to the results of the upper gastrointestinal endoscopy, a serum IgG4 test, and histopathological and immunohistochemical staining, he was finally diagnosed with IgG4-related oesophageal disease combined with a cytomegalovirus infection. We hope that through this case, we can learn more about IgG4-RD and, at the same time, give clinicians a better understanding of IgG4-RD combined with oesophageal ulceration, a new understanding of cytomegalovirus infections, and improved clinical knowledge.

## 1. Introduction

IgG4-related disease is a specific type of chronic autoimmune disease. The symptoms of IgG4-related disease are diverse and complex, and they are often accompanied by elevated serum IgG4 concentrations, with typical clinicopathological features, including dense infiltration of lymphocytes/plasma cells and fibrosis, large numbers of IgG4-positive plasma cells in the affected organs, storiform fibrosis, and obliterative phlebitis, which are reported to be the unique and characteristic features of IgG4-RD [1,2,3]. The disease can affect virtually any organ. The pancreas, biliary tract, salivary glands, lymph nodes, thyroid, kidney, lung, skin, gastrointestinal tract, prostate, and aorta are the most frequently involved organs, and one or more organs can be involved [2,4,5]. It is generally believed that immunity disorders and infections are probably the inductive factors of IgG4-RD. Della-Torre hypothesised that the pathogenesis of IgG4-RD resulted from the effect of T cells and B cells on certain specific antigens [6]. CD4+ cytotoxic T lymphocytes (CTL) express SLAMF7, perforin, granzyme, IL-1, TGF, and interferon γ, which may be important contributors to chronic inflammation and fibrosis in the pathology of IgG4-RD [7]. Regulatory T cells (Treg) in IgG4-RD lesions produce IL-10 and TGF-β. IL-10 is associated with antibody switching in B cells, and TGF-β is associated with tissue fibrosis [8]. The precise details of the mechanism of IgG4-RD remain unclear, and these remain to be investigated. Glucocorticoids and rituximab are the most established treatment options for the disease. If treatment is initiated early, substantial organ damage can be avoided. As IgG4-RD is a recurring disease, it needs to be monitored regularly, alongside active treatment. 

Cytomegalovirus (CMV) is a double-stranded DNA virus and is an important member of the herpesvirus family [9], which usually arises in patients with immunodeficiency or immunosuppression, and is generally infrequent in immunocompetent patients. However, the number of case reports and case series describing CMV infections in immunocompetent patients has been increasing [9,10,11,12]. Old age, critical illness, diabetes mellitus, chronic kidney disease, end-stage renal disease, and other comorbidities can lead to immune deficiency and increase the risk of CMV diseases [13,14,15]. CMV oesophagitis in healthy hosts has been reported, despite the pathological mechanisms still being unclear. Ganciclovir and valganciclovir are the main drugs currently used for treating CMV. Letermovir, a selective terminase inhibitor, is a new anti-CMV drug that inhibits the formation and release of viral particles and has a lower rate of myelotoxicity. It was previously approved for prophylaxis in hemopoietic stem cell transplant (HSCT) patients [16]. A review of the literature showed that IgG4-related oesophageal disease with cytomegalovirus infection is very rare.

The aim of this study was to give clinicians more understanding of IgG4-RD combined oesophageal ulceration, a new understanding of cytomegalovirus infection, and improved clinical knowledge. The study also aimed to enrich the clinical data and provide a reference for improving the early diagnosis and prognosis of this disease, and it might offer a reference for treatment.

## 2. Case Presentation

A 53-year-old male from Shanghai’s Song Jiang district in China had experienced nausea and vomiting and anaemia repeatedly for many years before this presentation. He had no previous medical history and no other autoimmune diseases. The patient had been diagnosed with eosinophilic gastroenteritis many years prior, and he made numerous hospital visits during the following years. The patient underwent abdominal magnetic resonance imaging and ultrasonography of the parotid gland in addition to gastric endoscopy at the other hospital, according to the medical records provided by the patient. The endoscopic findings showed a huge oesophageal ulcer. An oesophageal biopsy was performed, but no definitive diagnosis was made. Because of the symptoms recurrence, the patient came to our hospital for further treatment.

In September 2021, the patient presented to the gastroenterology department of our hospital with recurrent symptoms accompanied by dizziness and fatigue. The patient showed a poor general condition and an anaemic appearance. The rest of the physical examination was unremarkable.

To identify the cause, comprehensive laboratory tests were carried out on the patient, including a routine blood examination, stool tests, renal and liver function tests, human immunodeficiency virus, serum IgG4, immunological tests, CMV IgG and IgM, lymphocyte subsets, CMV DNA, TSPOT, and tumour markers. His urine, renal, and liver function tests were normal; the results of the other laboratory tests are shown in Table 1.

The chest and abdomen contrast-enhanced CT demonstrated wall thickening in the lower oesophagus with enlarged retroperitoneal lymph nodes and a small amount of ascites. An abdominal CT scan of the pancreas was performed and showed normal results (Figure 1).

Gastric endoscopy revealed a huge oesophageal ulcer, atrophic gastritis, gastric mucosal erosions, and a duodenal ulcer with lumen stenosis (Figure 2). The pathological examination of the oesophagus revealed predominant lymphoplasmacytic infiltration, extensive fibrosis, and inflammatory infiltrate. A malignant pathology was ruled out. Immunohistochemistry revealed IgG4-positive cells, with a ratio of IgG4-positive plasma cells to IgG-positive cells of >40%, and immunohistochemical staining using the anti-CMV monoclonal antibody was positive (Figure 3). The pathological examination of the gastric antrum and duodenal bulb revealed chronic inflammation of the mucosa.

Furthermore, his serum IgG4 levels were 4.05 g/L, validating his diagnosis of IgG4-RD with cytomegalovirus infection. The patient was treated with a proton pump inhibitor (PPI) and blood transfusions, and he was administered 0.25 g/q 12 h ganciclovir intravenously for 2 weeks. His haemoglobin was higher than before, and gastroscopy at follow-up after 2 weeks showed that the oesophageal ulcers had begun healing (Figure 4). He was discharged with a marked improvement in symptoms.

## 3. Discussion

In recent decades, IgG4-related disease has been recognised as a systemic disease with a fibrous inflammatory process associated with immunomodulation [17]. The precise prevalence of IgG4-RD is not known, as it is a relatively new entity and has been largely under-recognised in clinical practice [18]. The commonly affected organs involved include the pancreas, bile duct, major salivary glands, lacrimal glands, kidneys, lungs, and lymph nodes. Patients with IgG4-RD frequently present with multiple organ involvement, with a few isolated lesions. The oesophageal involvement of IgG4-RD (IgG4-RDE) has been uncommonly reported in the literature. Nese et al. [1] reported that only 15 cases in the literature have previously described the oesophageal involvement of IgG4-RDE. CMV-associated gastrointestinal diseases usually occur in immunocompromised patients; however, a few cases have also been described in healthy hosts [19]. As cases of IgG4-related isolated oesophageal lesions with cytomegalovirus infection are rare, it is easily misdiagnosed, especially as the disease has hidden and diversified clinical manifestations.

The clinical symptoms of IgG4-RD are dependent on the affected organs [20]. Previous reports indicated that IgG4-RDE is more frequent in males and occurs most often in elderly patients. The clinical symptoms of IgG4-RDE primarily include dysphagia and weight loss, and, more rarely, odynophagia, epigastric pain, and acid reflux [1]. Gastric endoscopies can reveal ulcerations and strictures, or mass-like lesions, mainly involving the distal and mid-oesophagus [21]. The typical manifestation of CMV oesophagitis is epigastric pain, fever, odynophagia, dysphagia, and gastrointestinal bleeding [10]. Endoscopic features can also be nonspecific, including erosions and ulcers of the oesophagus [19]. The endoscopic appearance of CMV oesophagitis ulcers is that of well-demarcated, vertical or horizontal, linear, shallow ulcers in the mid-to distal oesophagus [22]. In our case, the patient was a middle-aged male; the main clinical manifestations were repeated nausea and vomiting accompanied by anaemia. The CT scan revealed thickening of the wall of the oesophagus, and the pancreas was normal (Figure 1). Gastric endoscopy revealed a huge oesophageal ulcer (Figure 2). The clinical manifestations and CT scan lacked specificity, and there was no pathognomonic endoscopic feature for this patient; therefore, early diagnosis was difficult. In previous reports, the size of the IgG4-RDE mass in some cases ranged from 1.5 cm to 3.9 cm [23,24]. This case may be the largest IgG4-RDE-related ulcer in the literature.

Approximately 70% of patients with IgG4-RD have elevated serum IgG4 levels [25]. Serum IgG4 levels greater than 1.35g/L are used as the biomarker for IgG4-RD and have a diagnostic sensitivity ranging from 83% to 97% and a specificity of 60–85% [26]. In our patient, the serum IgG4 level was markedly elevated at 4.05 g/L (Table 1). CMV antigenemia is also known to be useful for the diagnosis of active CMV infection [27]. Anti-CMV IgM indicates a recent infection, and anti-IgG can be present in most adults who have previously been infected. Serology is not a gold standard for diagnosing CMV infections, and the absence of CMV-IgM antibody and antigenemia may not exclude CMV infections in an immunocompetent individual. In this case, the patient was negative for serum anti-CMV IgM, and his CMV-DNA titres, human immunodeficiency virus, and T-SPOT.TB results were all negative (Table 1). On the basis of the patient’s medical history, we were also able to rule out tuberculosis infection and human immunodeficiency virus, and there were no obvious signs of CMV infection in the serological tests. However, the patient was positive for the occult blood stool test, which may indicate upper gastrointestinal bleeding, and a decreased plasma albumin concentration (Table 1).

Because its clinical manifestations and endoscopic findings were atypical in the patient, without obvious clinical specificity, when IgG4-RD was suspected based on the elevated serum IgG4 concentrations, histopathology and immunohistochemical staining were carried out for a definite diagnosis of IgG4-RD. Histopathology and immunohistochemistry remain crucial to diagnosis. Typical histopathological features include storiform fibrosis, dense lymphoplasmacytic infiltrate, and obliterative phlebitis [18]. In our case, the stomach and duodenum were not involved; only the oesophagus was involved, presenting as an ulcerative oesophageal lesion. The pathological examination revealed predominant lymphoplasmacytic infiltration, extensive fibrosis, and inflammatory infiltrate; oesophageal malignancy was excluded. Immunohistochemistry revealed IgG4-positive cells, with a ratio of IgG4-positive plasma cells to IgG-positive cells of >40%, and, unexpectedly, revealed positive immunohistochemical results for CMV (Figure 3). On the basis of the 2020 revised comprehensive diagnostic criteria for IgG4-RD [28], our patient was diagnosed with IgG4-RDE. The standard approach for diagnosing CMV is an endoscopic biopsy of the inflammatory findings for histological confirmation of the intranuclear inclusions found through pathological examinations and positive CMV immunostaining [26]. Based on these findings, the diagnosis of IgG4-related oesophageal disease and CMV infection was clear.

IgG4-RD is considered a benign disease, and it needs to be differentiated from malignant tumours and similar benign conditions, such as primary sclerosing cholangitis, multicentric Castleman’s disease, secondary retroperitoneal fibrosis, granulomatosis with polyangiitis, sarcoidosis, and eosinophilic granulomatosis with polyangiitis. The detection of ANA, anti-dsDNA, anti-SSA, anti-SSB, anti-Ro-52, MPO-ANCA, and PR3-ANCA is valuable in differentiated diagnoses of IgG4-RD. However, in this case, these immunological tests were all negative (Table 1), and the endoscopic biopsy and CT findings showed no significant abnormalities; therefore, we ruled out malignant tumours, autoimmune disease, and connective tissue disease.

The difficulty of diagnosis also highlights the therapeutic challenges. This is the first case report of CMV oesophagitis encountered with IgG4-related oesophageal ulceration in immunocompetent patients. Thus, there is no relevant treatment experience to use for a reference. The first-line treatment strategy for IgG4-RD management consists of corticosteroid administration [29]. Ganciclovir, delivered by intravenous injection, was the first line of anti-CMV therapy, and the duration of the anti-CMV medication can vary from 3 to 16 weeks [30]. In this case, the patient was treated with ganciclovir by intravenous injection (0.25 g/q 12 h) together with a proton pump inhibitor for 2 weeks; steroids were not administered. Gastric endoscopy after the treatment revealed that the ulcers had healed (Figure 4). Our patient responded well to ganciclovir therapy, supporting the diagnosis of CMV infection.

Most studies on CMV oesophagitis have focused on patients with organ transplantations or HIV infection. We report a case of CMV oesophagitis in a non-immunodeficient patient. There are also no reports studying the association between IgG4-RD and CMV infection. It is worth exploring whether there is a relationship between IgG4-related disease and CMV infection, or whether this association was found simply by chance. As mentioned in previous reports, CMV reactivation in patients with oesophageal cancer has been reported [31]. IgG4-RD is considered to be an autoimmune disorder, and an immunocompromised status increases the risk of CMV infection or CMV reactivation. Therefore, this may be a possible explanation for the association between IgG4-RD and CMV infection.

## 4. Conclusions

IgG4-related oesophageal ulceration with CMV infection remains a particularly challenging diagnosis for physicians, mainly because of its rarity and the ease of misdiagnosis. For this reason, the key to diagnosis is histological examination. This case might provide a new visual guide for the aetiological diagnosis of oesophageal ulcers for gastroenterologists and will provide a reference for the appropriate therapy.

## Figures and Tables

**Figure 1 jpm-13-00493-f001:**
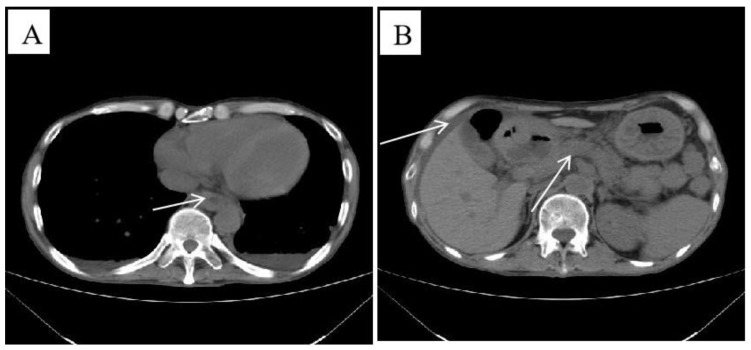
Chest and abdominal contrast-enhanced computed tomography (equipment type: Philips Brilliance CT 64 slice, Amsterdam, the Netherlands). (**A**) Image of the lower oesophageal wall, showing slight thickening (white arrow). (**B**) The pancreas appeared normal, with a small amount of ascites (white arrow).

**Figure 2 jpm-13-00493-f002:**
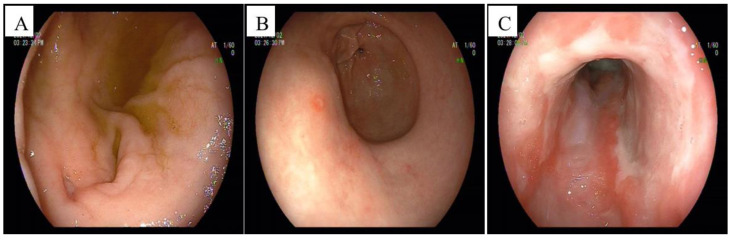
Initial findings of gastroscopy (equipment type: Pentax EG-2990i, Japan, Tokyo). (**A**) An ulcer (0.5 cm) in the anterior wall of the duodenal bulb, and the bulbar cavity showed oedema and stenosis. (**B**) Atrophic gastritis and gastric mucosal erosions. (**C**) The lower part of the oesophagus showed a huge oesophageal ulcer, between 37 cm and 45 cm from the incisors. The ulcer involved the oesophagus for about 4/5 weeks, and was covered with extensive white film.

**Figure 3 jpm-13-00493-f003:**
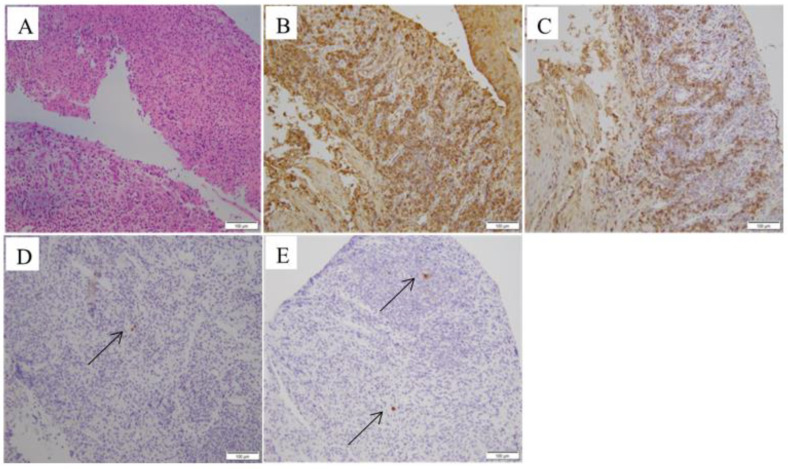
Pathological findings. (**A**). The oesophageal specimen revealed infiltration by lymphocytes, plasma cells, and eosinophils (H&E; ×100). (**B**). Immunoglobulin G (IgG) immunohistochemical staining showing increased numbers of IgG-positive plasma cells in the oesophagus lesions (HPF; ×100). (**C**) Dense infiltration of IgG4-positive plasma cells (HPF; ×100). (**D**,**E**) Immunohistochemical staining using anti-CMV monoclonal antibodies was positive (HPF; ×100) (black arrow).

**Figure 4 jpm-13-00493-f004:**
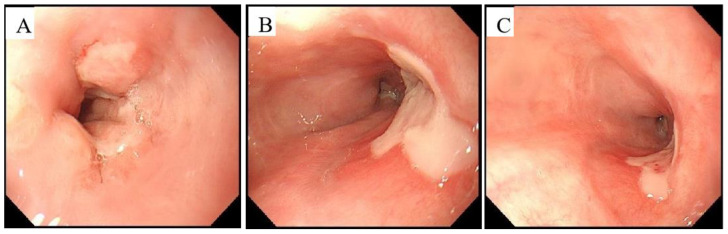
The findings of the second gastroscopy (after treatment) (equipment type: Olympus GIF-H260, Japan, Tokyo). (**A**) An ulcer (1.0 cm) in the dentate line. (**B**,**C**) An ulcer (2 × 1.0 cm) in the lower segment of the oesophagus about 40 to 42 cm from the incisors, and a 1.0 cm ulcer at the dentate line about 45 cm from the incisors, covered with white film at the bottom.

**Table 1 jpm-13-00493-t001:** Laboratory test results.

Laboratory Tests	Result	Reference Values
Haemoglobin	42 g/L	130–175 g/L
Plasma albumin	26.67 g/L	35–52 g/L
Serum IgG4	4.05 g/L	0–2 g/L
Anti-CMV IgG	500 U/mL	<1 U/mL
Anti-CMV IgM	0.26 COI	<1
CMV DNA	Negative	0–1000 copies/mL
Anti-HIV	0.189	<1
T-SPOT.TB	Negative	Negative
Lymphocyte counts	1167/μL	1530–3700/μL
Total T lymphocyte count	927/μL	723–2737/μL
B Lymphocyte count	50/μL	80–616/μL
NK cell count	198/μL	84–724/μL
Tumour mark	Normal	Normal
Stool occult blood test	Positive	Negative
anti-SSA	Negative	Negative
anti-SSB	Negative	Negative
anti-RO-52	Negative	Negative
anti-dsDNA	Negative	Negative
MPO-ANCA	Negative	Negative
PR3-ANCA	Negative	Negative

## Data Availability

The data presented in this study are available on request from the corresponding author.

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
