# Peer review of "IgG4-Related Oesophageal Disease with Cytomegalovirus Infection: A Case Report"

_jpm, 2023, doi:10.3390/jpm13030493_

Round 1

Reviewer 1 Report

IgG4 esophagitis with CMV infection  is avery rare condition, so the case report is interesting. Only several cases were  reported,  including [ Jang SW, Jeon MH, Shin HD. IgG4-Related Disease with Esophageal Involvement. Case Rep Gastroenterol. 2019 Sep 12;13(3):369-375. doi: 10.1159/000502794. PMID: 31607838; PMCID: PMC6787435]  – not cited in the article

Comments to the Case presentation

There is no need to duplicate numerical data in the text and table 1.

Figure 2. “the ulcer involved the esophagus for about 4/5 weeks – what does it mean?

 With regard to the differentia diagnosis of IgG4 RD – did you check for the exclusion criteria (such as anti-dsDNA, anti-SSA/Ro or SSB/La antibody and MPO- or PR3-ANCA.?)

As disease is rare, for educational purposes, I would recommend to cite and discuss more precise  the current diagnostic criteria [Umehara H, Okazaki K, Kawa S, Takahashi H, Goto H, Matsui S, Ishizaka N, Akamizu T, Sato Y, Kawano M; Research Program for Intractable Disease by the Ministry of Health, Labor and Welfare (MHLW) Japan.. The 2020 revised comprehensive diagnostic (RCD) criteria for IgG4-RD. Mod Rheumatol. 2021 May;31(3):529-533. doi: 10.1080/14397595.2020.1859710. Epub 2021 Jan 28. PMID: 33274670.]

 Please explain the term “ Storiform fibrosis”

Round 1

Response to Reviewer 1 Report

  1. IgG4 esophagitis with CMV infection  is avery rare condition, so the case report is interesting. Only several cases were  reported,  including [ Jang SW, Jeon MH, Shin HD. IgG4-Related Disease with Esophageal Involvement. Case Rep Gastroenterol. 2019 Sep 12;13(3):369-375. doi: 10.1159/000502794. PMID: 31607838; PMCID: PMC6787435]  – not cited in the article

Author Response

Thank you for your comments!

 I have read the literature you mentioned, the literature has been cited in the article.

Comments to the Case presentation

  1. There is no need to duplicate numerical data in the text and table 1.

Thank you for your comments!

duplicate numerical data in the text has been modified.

  1. Figure 2. “the ulcer involved the esophagus for about 4/5 weeks – what does it mean?

Thank you for your comments!

In previous reports, the size of the IgG4-RDE mass in cases with one ranged from 1.5 cm to 3.9 cm, However, larger, even 9cm has been reported, it means according to the endoscopic findings, the patient in this case was similar to malignant tumors. It needs to be differentiated from esophageal cancer.

  1. With regard to the differentia diagnosis of IgG4 RD – did you check for the exclusion criteria (such as anti-dsDNA, anti-SSA/Ro or SSB/La antibody and MPO- or PR3-ANCA.?)

Thank you for your comments!

  In this case, these immunological tests were all negative,I have added it to the article.

  1. As disease is rare, for educational purposes, I would recommend to cite and discuss more precise  the current diagnostic criteria [Umehara H, Okazaki K, Kawa S, Takahashi H, Goto H, Matsui S, Ishizaka N, Akamizu T, Sato Y, Kawano M; Research Program for Intractable Disease by the Ministry of Health, Labor and Welfare (MHLW) Japan.. The 2020 revised comprehensive diagnostic (RCD) criteria for IgG4-RD. Mod Rheumatol. 2021 May;31(3):529-533. doi: 10.1080/14397595.2020.1859710. Epub 2021 Jan 28. PMID: 33274670.]

Thank you for your comments!

 I have read the literature you mentioned, the literature has been cited in the article.

  1. Please explain the term “ Storiform fibrosis”

 However,elevated serum IgG4 concentration is frequently observed, and pathological examination usually reveals marked infiltration of IgG4-positive plasma cells with occasional fibrosis, a characteristic called storiform fibrosis. Storiform fibrosis is defined as spindle-shaped cells, inflammatory cells and fine collagen fibers forming a flowing arrangement.   I quote  from “The 2020 revised comprehensive diagnostic (RCD) criteria for IgG4-RD. Mod Rheumatol. 2021 May;31(3):529-533”.

Reviewer 2 Report

The article is well prepared. Studying interesting cases contributes to expanding knowledge and improving diagnoses. Case reports are a valuable source of knowledge for clinicians and significantly contribute to expanding their knowledge. The description of the case is accurate, but there is no information about the equipment on which laboratory tests, tomography and gastroscopy were performed. There is no scale in the photographs (figure 2), the photographs are correctly described.

The discuused case could be helpful to  learn more about IgG4-RD, and make clinicians have more understanding of IgG4-RD merging esophageal ulceration, and have a new understanding of cytomegalovirus infection and improve clinical knowledge.

Round 1

Response to Reviewer 2 Report

The article is well prepared. Studying interesting cases contributes to expanding knowledge and improving diagnoses. Case reports are a valuable source of knowledge for clinicians and significantly contribute to expanding their knowledge. The description of the case is accurate, but there is no information about the equipment on which laboratory tests, tomography and gastroscopy were performed. There is no scale in the photographs (figure 2), the photographs are correctly described.

The discuused case could be helpful to  learn more about IgG4-RD, and make clinicians have more understanding of IgG4-RD merging esophageal ulceration, and have a new understanding of cytomegalovirus infection and improve clinical knowledge.

Author Response

Thank you very much for your advice. I have added the  equipment about laboratory tests, tomography and gastroscopy to this article. The photographs (figure 2) were gastroscopy picture, there is no scale.